# Human movement and environmental barriers shape the emergence of dengue

Vinyas Harish [1,2,3], Felipe J. Colón-González [4,5,6], Filipe R. R. Moreira[7,8], Rory Gibb[4,5,6,9], Moritz U. G. Kraemer[10], Megan Davis [11], Robert C. Reiner Jr. [12,13], David M. Pigott [12,13], T. Alex Perkins [14,15], Daniel J. Weiss[16,17], Isaac I. Bogoch[1,18], Gonzalo Vazquez-Prokopec [19], Pablo Manrique Saide[20], Gerson L. Barbosa [21], Ester C. Sabino[22], Kamran Khan[1,2,11,23,24], Nuno R. Faria [8,22], Simon I. Hay [12,13], Fabián Correa-Morales [25], Francisco Chiaravalloti-Neto [26] & Oliver J. Brady [4,5,6] ✉

Understanding how emerging infectious diseases spread within and between countries is essential to contain future pandemics. Spread to new areas requires connectivity between one or more sources and a suitable local environment, but how these two factors interact at different stages of disease emergence remains largely unknown. Further, no analytical framework exists to examine their roles. Here we develop a dynamic modelling approach for infectious diseases that explicitly models both connectivity via human movement and environmental suitability interactions. We apply it to better understand recently observed (1995-2019) patterns as well as predict past unobserved (1983-2000) and future (2020-2039) spread of dengue in Mexico and Brazil. We find that these models can accurately reconstruct long-term spread pathways, determine historical origins, and identify specific routes of invasion. We find early dengue invasion is more heavily influenced by environmental factors, resulting in patchy non-contiguous spread, while short and long-distance connectivity becomes more important in later stages. Our results have immediate practical applications for forecasting and containing the spread of dengue and emergence of new serotypes. Given current and future trends in human mobility, climate, and zoonotic spillover, understanding the interplay between connectivity and environmental suitability will be increasingly necessary to contain emerging and re-emerging pathogens.

The geographic containment of emerging infectious diseases (EIDs) is a cornerstone of pandemic prevention and the basis of global agreements, including the International Health Regulations[1], that emphasise the prevention of national and international disease spread[2]. Despite this, little is known about what factors contribute to the early spread of EIDs. Outbreaks may end quickly due to stuttering transmission or successful containment, resulting in limited empirical data. Moreover, surveillance and diagnostics may not be sufficiently developed to accurately measure dispersal in the early phases of the outbreak[3,4]. Dengue is a vector-borne EID that has gradually expanded to over 120 countries since the 1940s with nearly 4 billion people now at risk[5,6]. Dengue spread is uniquely well-documented across Central and South America due to a continent-wide *Aedes aegypti* eradication programme that established early

surveillance systems across a diverse range of eco-epidemiological settings and delayed widespread dengue virus (DENV) transmission until the late 1980s[7]. The ultimate failure of this programme to contain dengue expansion created a unique case study on which to better understand modern infectious disease emergence.

Understanding how connectivity (mobility between invaded and non-invaded areas) and environmental factors shape routes of emergence could enable the prediction of future spread patterns. Implementing mosquito control in at-risk but not-yet invaded areas could contain the geographic expansion of dengue, emergence of new DENV serotypes[8], and enable early response to novel and re-emerging arboviruses transmitted by a common vector, such as Zika, chikungunya and yellow fever. Interest in containment has only become more acute as dengue's global burden has ballooned[9], recent serious Zika[10] and yellow fever[11] outbreaks have attracted global attention, and new dengue vaccines[12] and *Wolbachia* mosquito replacement technologies[13] offer hope of augmenting historically poorly effective and environmentally problematic vector control options[14].

Few countries have been able to reliably measure the expansion of dengue over decadal timescales[15,16], with limited historical data available for most other settings[8,17]. Predictive models that characterise the relationships between dengue spread and its known drivers such as temperature, rainfall, and connectivity offer the best chance of inferring generalisable mechanisms from limited data and allow useful insights for future containment strategies. While dynamic models of single outbreaks have incorporated the dual effects of mobility and environmental drivers[18], current frameworks for modelling long-term EID geographic spread have focussed on either connectivity[19] or environmental factors[20], despite the spread of species and diseases relying on a close interaction of connectivity and environmental suitability[21,22] for dispersal. Environmental factors may play a greater role in directing early spread if the pathogen is already circulating among highly connected areas, while connectivity may become more important in the later stages of spread as marginally suitable areas require repeated introduction to trigger an outbreak[23]. Modern sequencing techniques have allowed the geographic expansion of contemporary outbreaks to be reconstructed using phylogenetic methods[24], but sparse historical sampling limits our ability to infer subnational patterns, particularly pre-2000.

Here, we develop a dynamic modelling framework that integrates a wide range of environmental and human mobility-based features. We validate its ability to make predictions of dengue spread using data from 8026 municipalities over a 25-year timeframe across Mexico and Brazil, two of the highest dengue burden and most eco-epidemiologically diverse countries in Latin America. We then combine these models with phylogenetic analyses and historical outbreak records to test candidate origins of dengue in Brazil in the 1980s and, with the addition of climate change projections, to predict which areas in both countries are likely to be at risk up to 2039−demonstrating how our framework can characterise emergence and identify high-impact areas where interventions could limit spread.

## Results
### Observed patterns of spread 1996−2020
Mexico and Brazil have both observed substantial geographic expansion of dengue since the establishment of their national dengue surveillance programmes (Fig. 1). Here we define invasion as a total annual incidence above a country-specific threshold of ≥2 cases per 100,000 residents per year in Mexico and ≥20 cases per 100,000 residents in Brazil. These thresholds optimised the balance between identifying areas that, once invaded, regularly report cases (and can thus seed onward spread) with maximising the number of observed invasion events (allowing more detailed patterns of spread to be resolved). At these thresholds, we find that invaded municipalities report cases in 84% of post-invasion years in Brazil and 65% of post-invasion years in

Mexico, reflecting differences in the higher prevalence of low transmission, epidemic areas in Mexico (Supplementary Fig. 1).

In 1996, only 16 municipalities (0.65%) in Mexico met our definition of invaded and were spread across nine states in the southern part of the country with the biggest concentration in the eastern Pacific coastal state of Veracruz. The initial spread was minimal, but between 2000 and 2010, 965 municipalities were invaded with spread up the western and then eastern coastlines until the spread slowed into the 2010s (Fig. 1A, B). By the end of 2019, 1350 of 2456 municipalities nationally (55.0%) had exceeded our threshold for invasion.

In Brazil, dengue was present in all but two states (Rio Grande do Sul and Santa Catarina) and the Federal District (Distrito Federal) by the time national dengue surveillance was established in 2001. Since then, the number of total invaded municipalities has steadily grown from 549 (9.96%) in 2001 to 4229 (76.8%) in 2019 (Fig. 1C, D). Only isolated regions of the western and northern Amazon and southern states were defined as not having established DENV transmission by the end of 2019. Consistent with our expectation, patterns of expansion in both countries have been complex and spatially heterogeneous with spread from multiple sources that do not follow simple diffusion or smooth climatological gradients (Fig. 1A, C).

### Modelling spread 1996−2020
To reconstruct, understand and project these complex patterns of spread we developed a temporally-dynamic, geospatial modelling approach. Our two-fold approach first uses a hierarchical survival (temporal) model to predict the total number of municipalities invaded each year without any connectivity or environmental covariates. Second, a machine-learning (geospatial) model trained on year-on-year changes in invasion sources and a range of environmental and connectivity features determines the spatial distribution of invaded municipalities each year ("Methods" section).

Despite important heterogeneities, the annual total number of invaded municipalities was well represented by covariate-free parametric survival models that gave better predictive performance than more flexible spline-based approaches (Supplementary Fig. 2A, B). Furthermore, our expanding window timeseries cross-validation fitting showed that both the functional form and parameterisation (trajectory) of these survival models coalesce relatively quickly after 4−5 years of fitting data are available (Supplementary Fig. 2B and D). This suggests that the long-term total number of invaded municipalities each year is relatively predictable and that the timing of saturation can be estimated even early on in the invasion process.

By fitting a machine-learning model, we showed that the spatial distribution of which municipalities were invaded each year could also be characterised and predicted (Fig. 2, Supplementary videos 1 and 2). Consistent with previous disease mapping studies[25,26], we found that increasing geospatial model complexity was necessary to capture the non-linear and interacting influence of climate and connectivity features. Gradient-boosted decision trees (GBDT) were found to be the most optimal method and gave substantial improvements over a simple logistic regression (Area Under the Curve [AUC] Mexico: 0.87 vs 0.94 and Brazil: 0.75 vs 0.88, Supplementary Fig. 3) when evaluated using a simple random train-test split of the year-to-year spread data ("Naïve model"). When this spread model was initialised with the observed invaded areas at the beginning of the timeseries (1996 Mexico, 2001 Brazil) and then simulated with an annual timestep up to the year 2019 ("Simulation model") it could predict observed invasion date within ± 2 years for 75% of municipalities in Mexico and 81% in Brazil with even performance across sub-national regions (Supplementary Fig. 4A, B). However, predicting which municipalities would be invaded over the next calendar year (i.e. as would be required for forecasting) was more challenging with an expanding window timeseries cross-validation ("Hindcast model") showing low sensitivity, particularly in Mexico (mean 0.13 and 0.21 for Mexico and Brazil

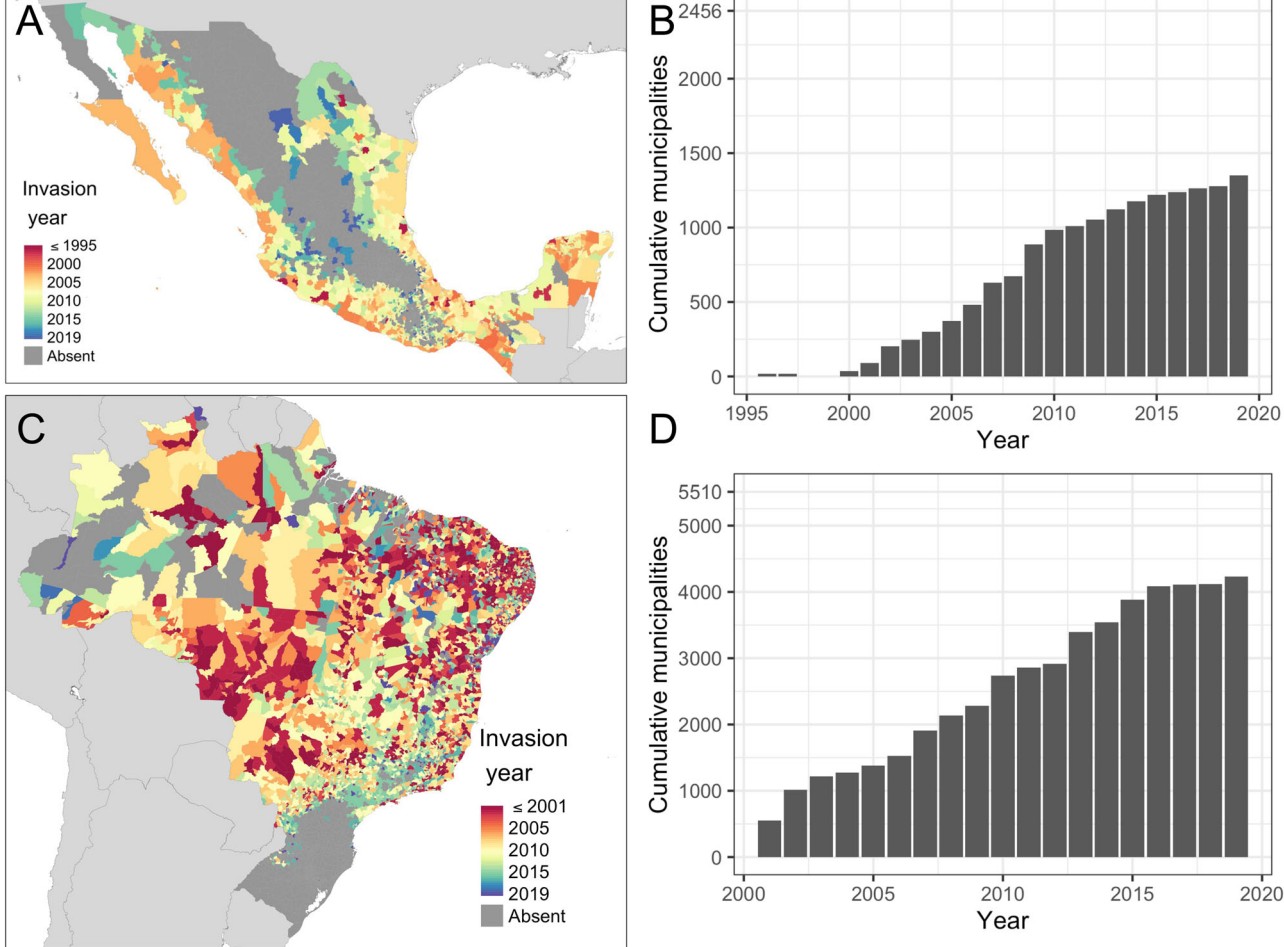

**Fig. 1 | Spatiotemporal distribution of reported dengue-invaded municipalities (Mexico: 1995-2019, Brazil 2001–2019).** Distribution of reported dengue-invaded municipalities over space (**A**, **C**) and time (**B**, **D**) for Mexico (**A**, **B**) and Brazil (**C**, **D**). A municipality is defined as invaded in the year it first exceeds an optimised cumulative incidence threshold (2 cases per 100,000 residents in Mexico, 20 cases per 100,000 residents Brazil, see "Methods" section). Source of Administrative boundaries: The Global Administrative Unit Layers (GAUL) dataset, implemented by FAO within the CountrySTAT and Agricultural Market Information System (AMIS) projects. Source data are provided as a Source Data file.

respectively, Supplementary Fig. 4C, D). This suggests that higher volumes of data are required to accurately predict the spatial pattern of spread and that while year-on-year patterns of spread can be stochastic, the longer-term spread trend may be more deterministic and predictable.

Despite the strong predictive performance of the combined temporal and geospatial model ("spread model"), some locations were consistently difficult to predict. For Mexico, the model under-predicted the rate of spread in southern parts of the country and in the Yucatan peninsula and over-predicted spread in some coastal cities (Fig. 2C). This may be due to our model underpredicting the relatively higher spread potential of sources in the year-round southern tropics. In Brazil, the model over-predicted the uniformity of the spread of dengue into the country's interior which showed more heterogeneity in observed values than predicted (Figs. 1C and 2D, F), possibly suggesting the involvement of more fine-scale mobility patterns in these regions.

A range of environmental and connectivity features predicted invasion risk (Fig. 3). Nighttime temperature and connectivity metrics based on municipality adjacency, long-term migration patterns, and radiation movement models were among the most consistently important features in both countries with higher values conferring greater invasion risk (Fig. 3A, C, Supplementary Fig. 5). This is consistent with the known constraints temperature places on the

altitudinal and latitudinal limits of *Ae. aegypti* mosquitoes[27] and the role frequent, commuter-style human movement plays in spreading dengue between peri-urban and urban environments[19,28,29]. Despite the climatological and epidemiological differences between the two countries, the spread model selected features and effects were similar. Models trained on Brazilian data and used to predict dengue spread in Mexico, and vice versa, only resulting in minor drops in predictive performance (Supplementary Fig. 6), suggesting the process of dengue spread could generalise and could be used in areas where dengue is currently emerging (e.g., Argentina and Chile).

Overall, the combined invasion risk of all environmental features outweighed the combined risk from connectivity features for 56% of invasion events in Mexico and 57% in Brazil but importance changes of time and space were observed. Environmental features contributed more to invasion risk in the early years before connectivity features became more predominant from ~2010 onwards in both countries as dengue became more widespread (Fig. 3B, E). Connectivity features tended to contribute more to invasion risk in large cities and sparsely populated areas (Fig. 3C, F). Combined these results suggest that, like the spread of invasive species[22], the early spatial spread of dengue is shaped by environmental factors that allow a spreading epidemic to establish in certain areas, resulting in a patchy but broad distribution (Fig. 1A, C). As more areas become invaded, connectivity becomes more important to increase the frequency of introduction to isolated

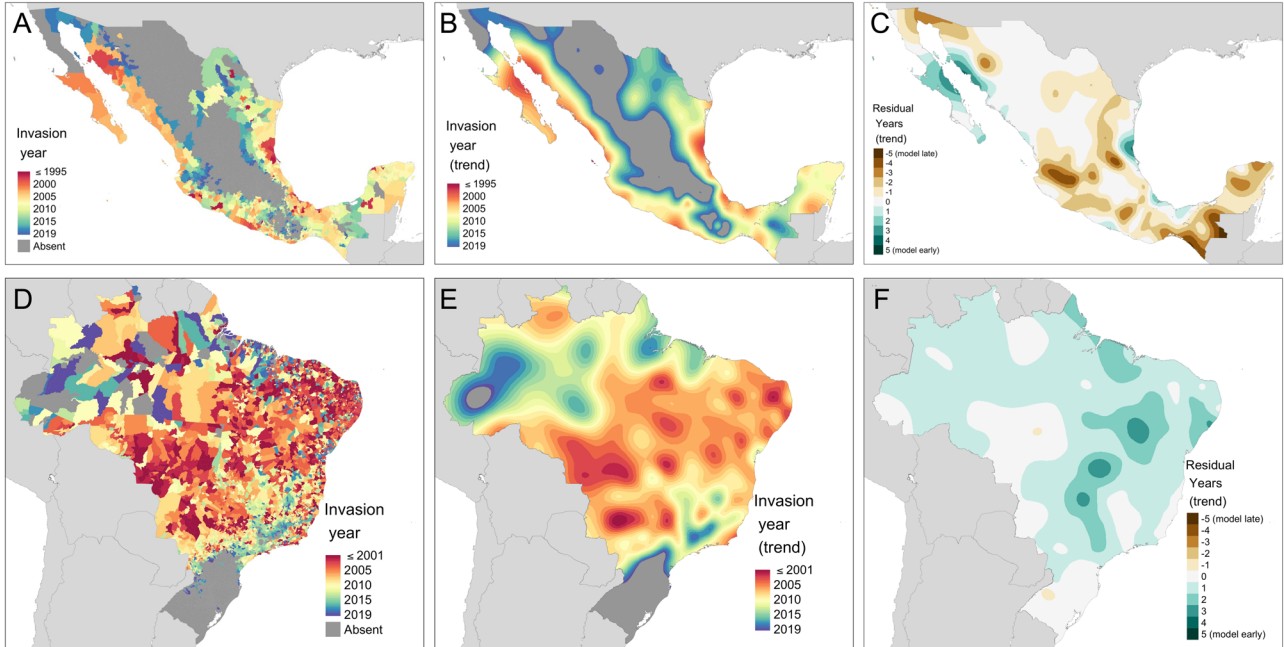

**Fig. 2 | Predicted year of dengue invasion.** Predicted year of invasion for Mexico (**A–C**) and Brazil (**D–F**) since the beginning of national dengue surveillance (Mexico: 1995–2019, Brazil 2001–2019). **A, D** give raw municipality-level predictions while **B, E** summarise spread trends using thin-plate splines. **C, F** show smoothed trends of model residuals where brown colours show areas where dengue was reported before predicted (or never predicted- assigned the value −5) by the model and green vice versa. Source of Administrative boundaries: The Global Administrative Unit Layers (GAUL) dataset, implemented by FAO within the CountrySTAT and Agricultural Market Information System (AMIS) projects. Source data are provided as a Source Data file.

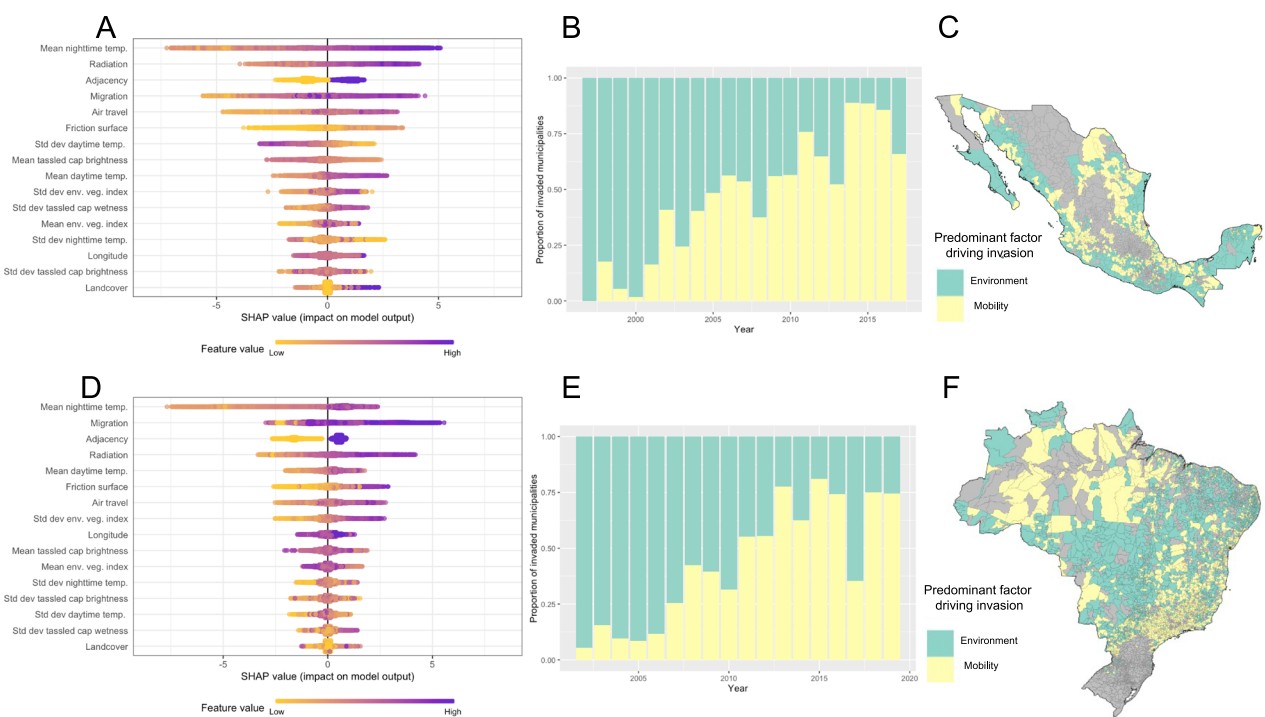

**Fig. 3 | Feature importance summaries.** Feature importance of gradient-boosted decision tree models for Mexico (**A**) and Brazil (**D**) using Shapley value summary plots. Colours indicate the relative values of each feature (rows) and their impact on model prediction (positive values = increased invasion risk). Features are ordered by mean impact on model output with the top variable conferring the most impact. Temp. temperature, Std dev standard deviation, env. veg. index environmental vegetation index. Timeseries plots and maps show if the contribution to the model-predicted invasion risk is greater for all environmental features or all mobility features for the year in which each municipality was invaded for Mexico (**B**, **C**) and Brazil (**E**, **F**). Source of Administrative boundaries: The Global Administrative Unit Layers (GAUL) dataset, implemented by FAO within the CountrySTAT and Agricultural Market Information System (AMIS) projects. Source data are provided as a Source Data file.

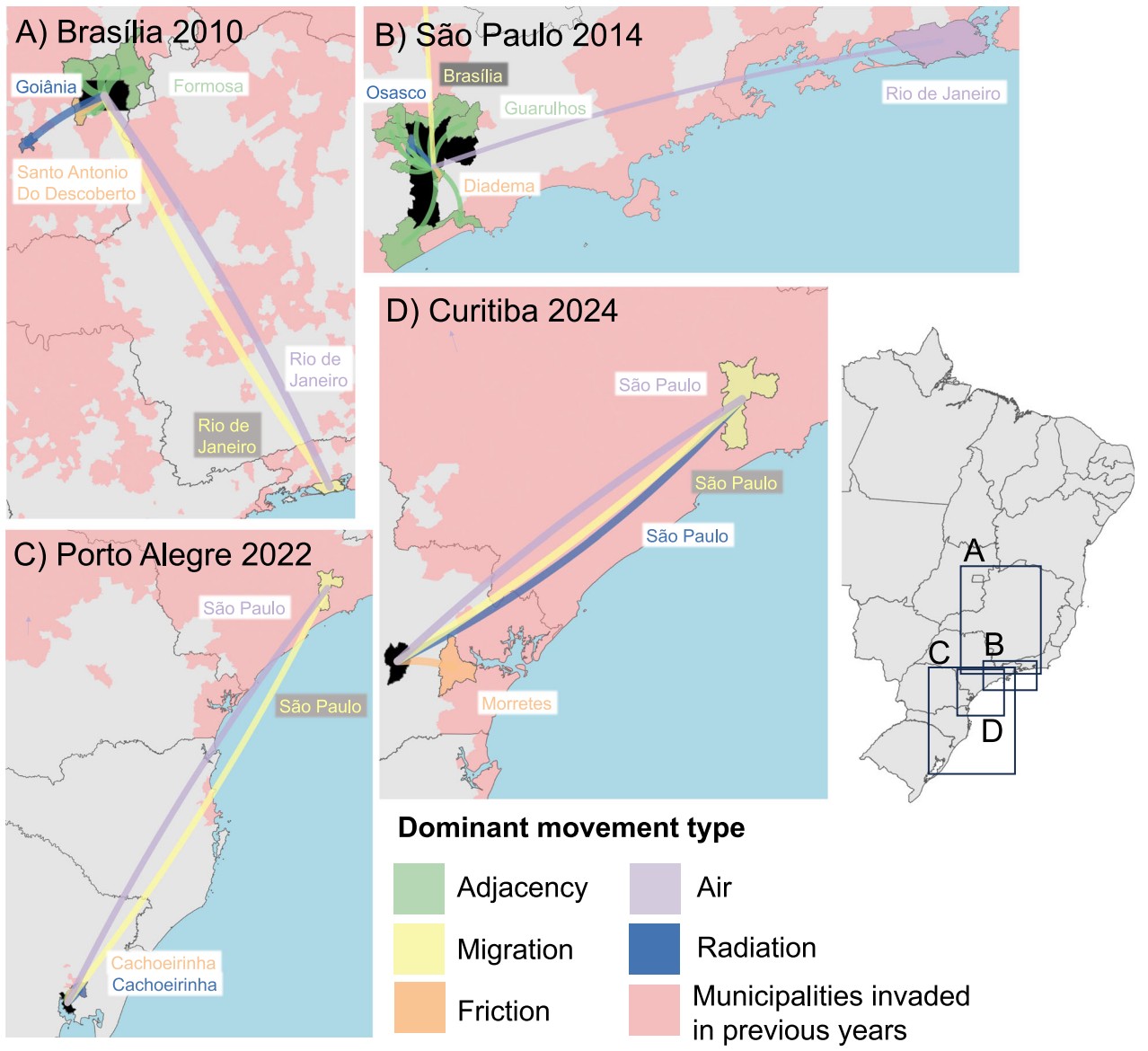

**Fig. 4 | Routes of dengue importation in major municipalities, Brazil.** Predicted routes of dengue importation for select large cities in the past (**A**, B) and future (**C**, D) in Brazil. Maps show all previously invaded municipalities (red) and the most connected previously invaded municipality for each different type of human movement included in the spread model for the year in which each city was (or is predicted to be) invaded. The invaded municipality is shaded in black. Source of Administrative boundaries: The Global Administrative Unit Layers (GAUL) dataset, implemented by FAO within the CountrySTAT and Agricultural Market Information System (AMIS) projects. Source data are provided as a Source Data file.

areas or areas that may only be environmentally suitable at certain times.

### Mapping routes of spread

A closer examination of the mobility networks associated with the observed invasion of the largest cities (Figs. 4 and 5) suggests that invasion is a multi-stage process. For Monterrey, Guadalajara, Brasília, and São Paulo, multiple lower-density neighbouring areas or nearby cities were invaded in the years before the city centre itself. In all cases, connectivity via long-distance air routes was present for many years before the invasion occurred. In Brazil, Rio de Janeiro (invaded in or before 2001) showed a high degree of air and migration connectivity with both Brasília and São Paulo but invasion did not occur until 2010 and 2014 respectively (Fig. 4A, B), with Cancún (invaded since 2001) playing a similar role in Mexico (Fig. 5A, B). This would suggest that for major cities to be invaded, connectivity by air is necessary (as also demonstrated by the geospatial model (Fig. 3A, D)) but not sufficient

for the invasion which instead requires the combined importation pressure of both nearby and longer distance links (Figs. 4 and 5). The finding is in contrast with observations from Thailand where epidemic waves were observed to spread through large city hubs in a general urban-to-rural direction[29,30], but may reflect differing distributions of the highest vulnerability areas within cities in Thailand, Mexico, and Brazil. Invasion route predictions are challenging to validate with prospective data collection (although modern phylogeographic methods offer some potential[31,32]). However, such predictions could be used to concentrate mosquito control in high-vulnerability areas within a city or in its suburbs—enabling more effective and efficient containment than the current, reactive methods.

### Reconstructing the origins of DENV spread in Brazil

By the time nationwide dengue surveillance was established in Brazil in the year 2000, DENV transmission was widespread (Fig. 1C), making it impossible to directly observe the early geographic origins of DENV

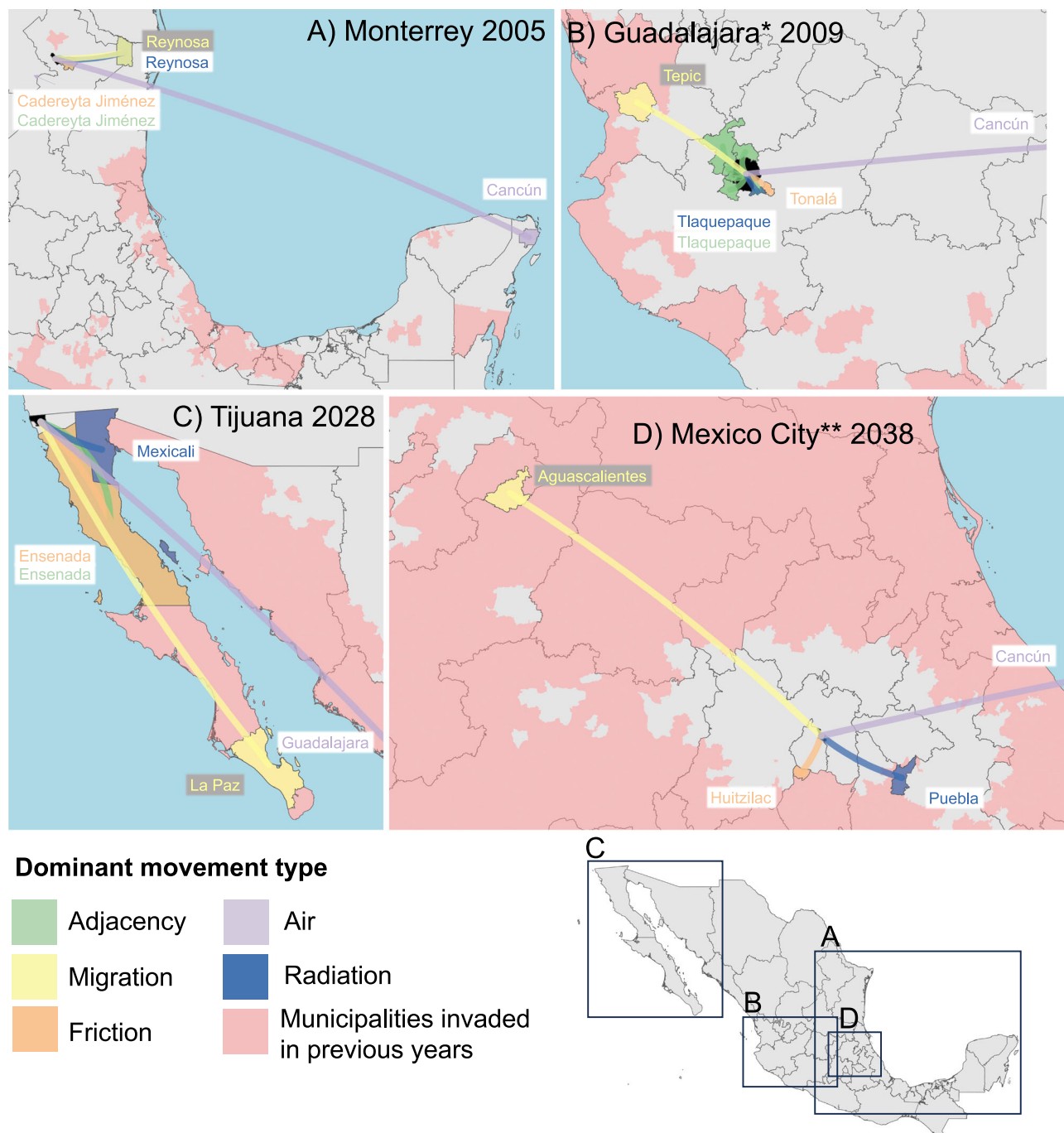

**Fig. 5 | Routes of dengue importation in major municipalities, Mexico.** Predicted routes of dengue importation for select large cities in the past (**A**, **B**) and future (**C**, **D**) in Mexico. Maps show all previously invaded municipalities (red) and the most connected previously invaded municipality for each different type of human movement included in the spread model for the year in which each city was (or is predicted to be) invaded. The invaded municipality is shaded in black. * Zapopan municipality. ** Nezahualcóyotl municipality. Source of Administrative boundaries: The Global Administrative Unit Layers (GAUL) dataset, implemented by FAO within the CountrySTAT and Agricultural Market Information System (AMIS) projects. Source data are provided as a Source Data file.

spread in Brazil, unlike in Mexico (Fig. 1A). Several possible sources have been suggested based on sporadic outbreak reports and phylogenetic analyses of DENV sequence data. Sporadic outbreak reports[17] suggest four potential geographically distinct introductions that preceded wider outbreak reports in their general vicinity: Rio de Janeiro (RJ) 1986, Fortaleza (CE) 1986, São Paulo (SP) 1990 and Manaus (AM) 1996 (Supplementary Fig. 7). While outbreaks were reported in Brazil prior to these dates, beginning with the 1981-82 Boa Vista outbreak, such outbreaks did not persist across multiple years, limiting their ability to seed wider spread[33,34]. Phylogenies reconstructed from DENV

sequences (Fig. 6 and Supplementary Table 5) suggest DENV serotype 1 (D1, genotype I, lineage BR1) was circulating in Rio de Janeiro state with a slightly earlier estimated date of arrival of 1983 (95% Credible Interval, CI 1982–1985). This analysis also estimates an independent introduction (D1, genotype I, lineage BR2) into northern Brazil in the late 1990s (Roraima state, assumed Boa Vista, 1998 CI 1996–1999, Fig. 3A). Unlike the sporadic outbreak reports, these phylogenies do not support an independent introduction to São Paulo state around 1990 and show no sustained lineages in the Northeast region until the mid-to-late 1990s (D1-BR1, D2-BR1 and D3-BR1, Fig. 3A).

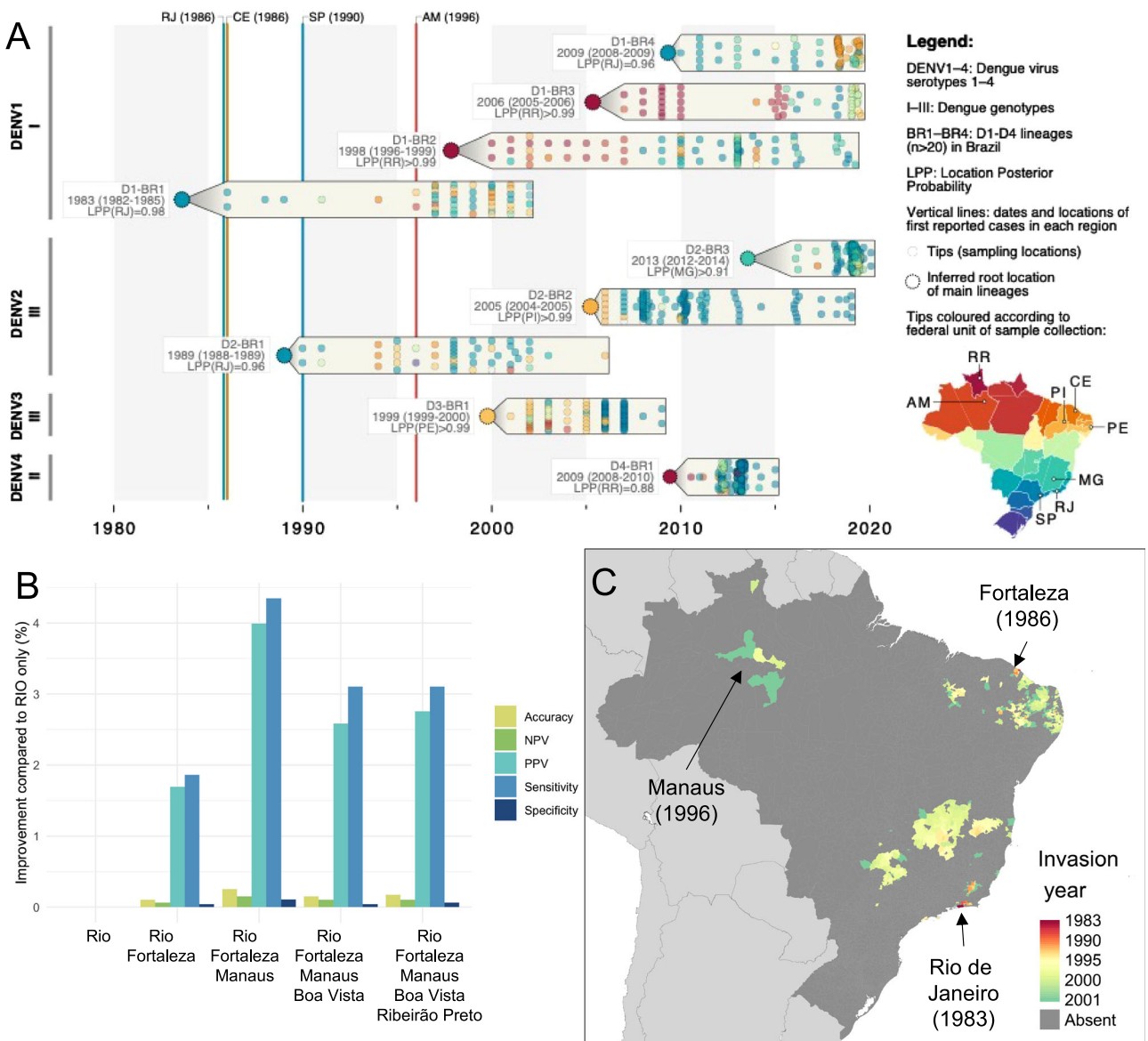

**Fig. 6 | Predicted historical expansion of dengue in Brazil (1983–2001).** Plausible origins of dengue introduction are identified from phylogenetic (horizontal clades) and sporadic outbreak reports (vertical lines) records in (**A**). The fit of the spread model to the observed distribution in the year 2001 when initiated with different combinations of these sources is shown in (**B**) (only best-performing model with 1–5 sources shown) with predictions of the best fitting model up to 2001 shown in (**C**). NPV negative predictive value, PPV positive predictive value. Source of Administrative boundaries: The Global Administrative Unit Layers (GAUL) dataset, implemented by FAO within the CountrySTAT and Agricultural Market Information System (AMIS) projects. Source data are provided as a Source Data file.

Assuming the same factors and processes that drove dengue spread 2001–2019 also acted similarly prior to 2001, we can use our spread model to predict the historical expansion of dengue in Brazil. We can also test the relative likelihood of different candidate sources of DENV introduction by comparing the predicted spread of dengue from the simulation model initialised with different sources with the observed distribution in 2001. Simulating spread 1983–2001 following a single introduction in Rio de Janeiro did lead to spread in multiple parts of the country, but with a bias towards cities on the East and North coasts and limited spread inland, in contrast with the observed distribution in 2001 (Fig. 1C). Adding an additional introduction to Fortaleza in 1986 improved the pattern in the Northeast, but recovering inland spread was only possible when a further introduction to Manaus in 1996 was included (Fig. 6B, C). Adding additional or different combinations of sources did not improve model fit to the observed distribution in 2001 (Fig. 6B), suggesting a limited number of

introductions are sufficient to explain the widespread rapid expansion of dengue in Brazil.

Consistent with phylogenetic analyses, our spread model suggests there was no sustained introduction to São Paulo around 1990 (as suggested by historical outbreak reports). However, our spread model does support an introduction in the Northeast in the mid-to-late 1980s, showing how our spread models can combine and overcome gaps in epidemiological and sequence data (e.g., we only found 26 DENV sequences in the Northeast region prior to 2000). Although we were able to reconstruct the broad pattern of historical invasion in Brazil (Supplementary video 2), we were unable to predict invasion to many smaller isolated inland municipalities leading to lower model sensitivity (accuracy 0.85, sensitivity 0.31, specificity 0.91) when evaluated on the observed distribution in 2001. While these invasions could have occurred due to further unobserved international introductions, these municipalities have no obvious connections outside

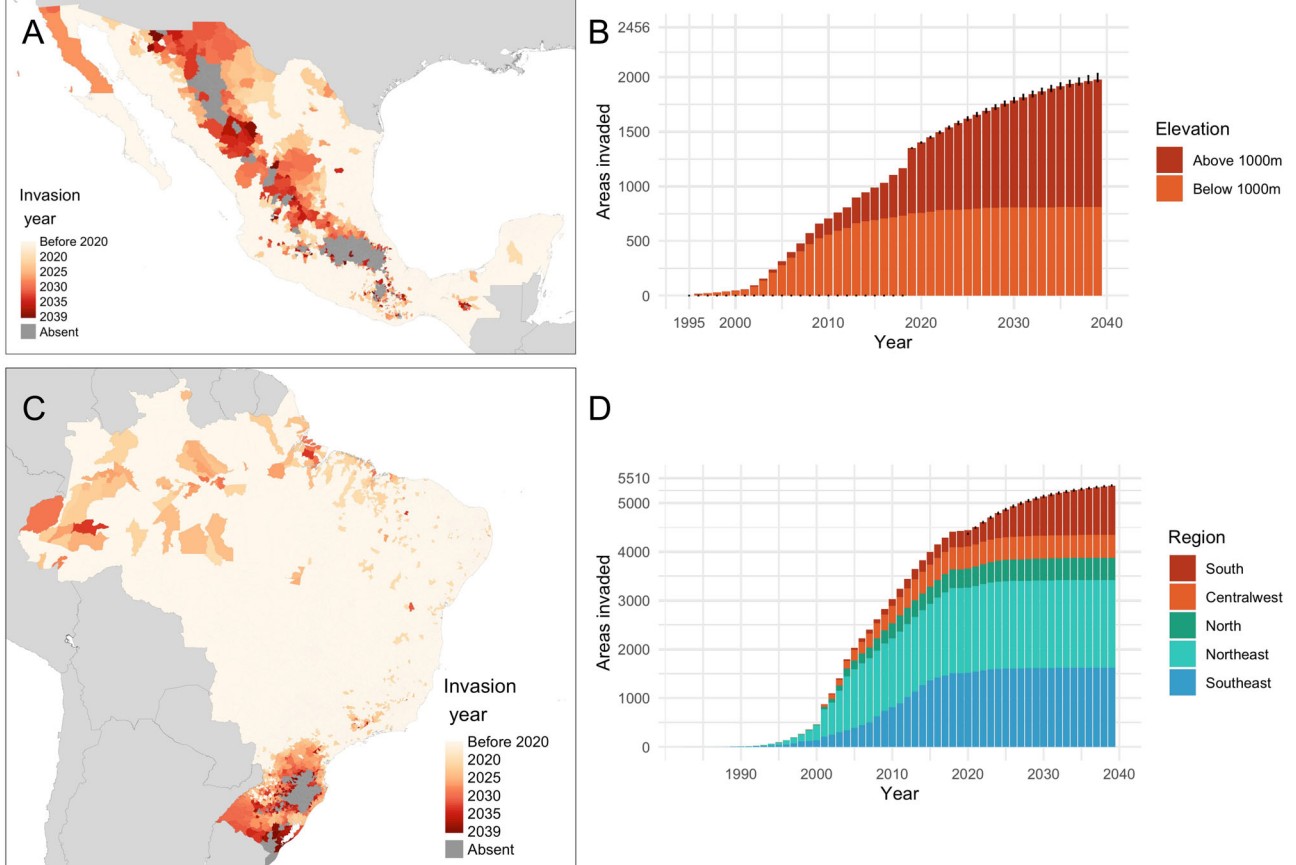

**Fig. 7 | Predicted future spread of dengue in Mexico and Brazil 2020–2039.**
**A**, **C** show the spatial distribution while **B** and **D** show the breakdown of invaded municipalities over time with respect to elevation in Mexico and geographic region in Brazil respectively. Error bars in **B** and **D** show the 95% credible intervals for the total number of municipalities invaded per year for each year from 2020 onwards based on an ensemble of five temporal survival models. Source of Administrative boundaries: The Global Administrative Unit Layers (GAUL) dataset, implemented by FAO within the CountrySTAT and Agricultural Market Information System (AMIS) projects. Source data are provided as a Source Data file.

Brazil and may instead suggest that other forms of human movement not accounted for in our models (e.g., multi-stop bus or truck traffic[35] or fluvial travel, particularly in the Amazon region) may have been more important for dengue spread prior to 2001.

More broadly, this independent evidence from DENV sequence data also served to validate two key assumptions of our dengue spread model. First, once introduced, DENV lineages persist for a long time (10+ years) and present a continual threat of exporting risk to new areas. Second, at least after 2001, introductions of novel DENV lineages occur (or are first detected) in areas that already have sustained DENV transmission, justifying the focus on domestic spread for these settings and suggesting human immune dynamics may affect suitability for the persistence of novel DENV genotypes[36] as has been observed in other longitudinal studies of DENV genotype replacement.

### Predicting future spread between 2020 and 2039

The simulation model was then used to project which of the remaining uninvaded areas are at risk of invasion between 2020 and 2039. Risk over this time period is determined by a continuation of the spread process, but also changing climatic and environmental factors that can encourage or limit transmission in different areas[6]. We account for changing environmental factors by projecting temperature, vegetation, and precipitation features based on the latest Coupled Model Intercomparison Project Phase 6 (CMIP6) study[37] (Supplementary Information 1.4 and Supplementary Fig. 8).

We predict that dengue will continue to undergo significant expansion between 2020 and 2039 with the percentage of municipalities invaded increasing from 76.8% to 97.2% (95%CI 97.0%–97.4%) in Brazil and 55.0%–81.5% (CI 80.4%–82.7%) in Mexico. In Mexico, the spread is predicted to be primarily inland into the high-altitude central plateau with 91.0% of future invasions municipalities with an average elevation over 1000 m (Fig. 7A, B, Supplementary videos 1 & 2). This will include spread to the last few remaining dengue-free large cities in Mexico with the areas around Tijuana, bordering the USA in the far north, expected to be invaded between 2027 and 2030 and the first invasions into metropolitan Mexico City between 2038 and 2039 (Fig. 5C, D). Invasion into Tijuana is expected as a continuation of a gradual spread up the Gulf of California with the highest invasion risk from the neighbouring municipalities of Ensenada and Mexicali as well as the more distance regional centres of La Paz and Guadalajara (Fig. 5C). The first invasions into the Mexico City metropolitan area are expected to occur when longer distance connectivity (Cancún) combines with links to other more proximal regional cities (Aguascalientes and Puebla, Fig. 5D). In Brazil, the majority (60.7%) of areas invaded 2020–2039 will be in the South region with the more isolated areas of the state of Santa Catarina and Rio Grande do Sul being the last to be invaded (Fig. 7C, D). With strong connections with São Paulo in combination with a gradual advance of dengue down the southern Brazilian coastline, the two biggest cities in southern Brazil (Porto Alegre and Curitiba) are expected to be invaded in 2022 (CI 2022–2022) and 2024 (CI 2024–2025) respectively (Fig. 4C, D). Despite frequent introductions, Porto Alegre rarely experienced sustained transmission and had a DENV seroprevalence of less than 1% as recently as 2015[28], however, in the first five months of 2022, the city

experienced its first major outbreak with 3850 notified cases[38] (27 cases per 10,000 residents), signalling the first year in which the area exceeded our threshold for invasion.

The pattern of invasion risk resulting from a combination of proximal and distal sources is expected to continue with the invasion of smaller close by cities pre-empting the arrival of dengue (Figs. 4C, D and 5C, D). This cumulative effect of importation pressure from multiple sources allows the last few remaining biogeographical barriers to be overcome.

These predictions suggest a more extensive and rapid expansion of dengue in Mexico and Brazil than previously thought. Previous predictions of future dengue risk are derived from models that consider environmental change, but not connectivity nor the non-linear, dynamic, dependent nature of the underlying spread process[6,39,40]. Environment-only, ecological niche models may over-attribute dengue absence to environmental features, particularly as the areas that are currently least environmentally suitable (e.g., dry, high altitude or higher/lower latitude) are, for now, poorly connected with areas of active transmission. This could be problematic for settings such as Mexico and Brazil where the distribution of dengue is rapidly expanding, exposing areas previously judged to be unsuitable to new levels of invasion pressure, especially given the finding that connectivity becomes increasingly important over time (Fig. 3B, E). The predictions from these spread models represent a "business as usual" scenario for dengue control efforts that are assumed to stay at current levels, and thus they could be used to simulate and optimise different containment strategies to limit future spread more effectively.

## Discussion

Our analyses showed that the expansion of dengue in Mexico and Brazil follows consistent and predictable pathways that are shaped by an interaction between the environmental suitability of the destination and connectivity with potential sources. By using models that account for these drivers we showed that the modern spread of dengue in Brazil could be explained by just three introductions to Rio de Janeiro, Fortaleza and Manaus between 1983 and 1996, identify likely proximal and distal routes of invasion for specific cities, and project the timing of future spread into highland regions of Mexico, including Mexico City, and southern Brazil. This represents the first time, to our knowledge, that spatial models of disease spread have informed origins, pathways, and future projections of an emerging infectious disease.

These maps and models can be used to develop early warning systems and containment strategies for dengue, related arboviral diseases and, with adaptation, other EIDs[41]. Our country cross-validation (Supplementary Fig. 6) suggests that our models could immediately be used to predict the spread of dengue in other countries, including those where emergence is in earlier stages, although re-fitting to at least five years of high-resolution data on spread would improve predictive performance (Supplementary Fig. 2). Another immediate application of these models could be to predict the spread of new DENV serotypes because serotype switching is commonly associated with dengue outbreaks and more severe disease outcomes[42,43]. Here we focus on the initial invasion of dengue, but comparisons between this model and models with more complex covariates for immunity based on past serotype prevalence could be used to test different hypotheses about how between-serotype dynamics affect the emergence of hyperendemicity. Due to shared vector species, our models could also give insights into the current emergence and re-emergence of Zika, chikungunya, and yellow fever in the Americas as well as suggest how coordinated and targeted interventions could contain the spread of these arboviruses to a more limited range than dengue. Interest in such targeted containment strategies has recently been boosted with successful trials of new interventions such as vaccines[12] and the release of *Wolbachia*-invaded mosquitoes[13]. Disruption to

domestic and international human movement in response to the COVID-19 pandemic revealed how dependent modern DENV transmission is on connectivity, with an unexpected near-global decline in incidence in 2020[44] and local elimination in some non-endemic areas[45]. Targeting mosquito control to high-risk spread routes at both origin and destination could achieve similar levels of containment and should be studied as a future arbovirus containment strategy. The modelling approach developed here could also be adapted to understand the spread of other EIDs by re-selecting relevant features and collecting high-resolution spatiotemporal data on disease spread. Seven of the eight WHO-listed priority emerging pathogens[46] require a vector to spread or depend on interaction with a non-human reservoir. Emergence risk will, therefore, depend on both environmental suitability and connectivity necessitating modelling frameworks such as that presented here. Adaptation of these frameworks to past outbreak data could improve estimates of locations and times of the initial unobserved zoonotic spillover which, in turn, could improve surveillance at the animal-human interface and pandemic preparedness[47].

Our results are subject to limitations with the data and model assumptions that may affect the applicability of some of the results. The proportion of DENV infections reported varies between and within countries and over time[48]. While we saw no evidence of national-level changes in surveillance policy increasing observed invasion rates (Fig. 1B, D), surveillance gaps and biases may explain some discrepancies between observed and predicted arrival times. We also assumed the continual presence of dengue in a municipality once it exceeds our invasion threshold. Changing immunity and the arrival of new serotypes will change the infectivity of a municipality over time, but previous studies in Southeast Asia have suggested that, despite these differences, transmission is surprisingly synchronous over broad geographic areas[49]. Further analysis on the spread of epidemic waves disaggregated by the magnitude of the epidemic in source locations and seasons would be of interest since we did not consider intra-year variations in spread processes[50]. Due to the limited availability and comparability of international dengue data sources[51], we were also unable to include international importation into our spread models and thus assume that geographic spread occurs primarily from domestic sources. While international importation has been shown to be an important driver of DENV serotype and lineage replacement[34], such introductions mostly occur (or are most commonly detected) in areas where DENV has already been circulating, often for many years, suggesting such model developments would be more important for predicting serotype spread than, as we do here, the first arrival of dengue.

In a time of unparalleled increases in human mobility, climate change, and zoonotic spillover it has never been more important to understand how these factors interact to shape EID emergence. Uncontained EIDs, including dengue, provide a valuable opportunity to understand disease spread with the ultimate goal of designing better strategies to contain future pandemics.

## Methods
### Dengue spread data
Dengue case data was obtained at the municipality level (2nd administrative level) for Brazil and Mexico. Municipalities had a median size of 355 square kilometres (interquartile range 161–898) and a median population in 2010 of 12,257 (interquartile range 5414–27,863). For Brazil, annual total dengue cases were extracted for 5570 municipalities for each year for which national data were available (January 2001–December 2019) from the Notifiable Diseases Information System (SINAN), obtained via the Ministry of Health Information Department (DATASUS)[15,52,53]. Total annual dengue cases included suspected and confirmed cases from a range of healthcare settings, and each case had an associated infection time defined as the month of first symptoms. For Mexico, annual total dengue cases for 2456 municipalities

were extracted for each year 1995–2019 from the Sistema de Vigilancia Epidemiológica de Dengue via the Instituto Nacional de Transparencia[16]. This database contains a mixture of case definitions with only Dengue Haemorrhagic Fever (DHF) reported 2000–2008, combined dengue and DHF 1995–1999 and 2009–2015, and combined non-severe dengue, dengue with warning signs and severe dengue 2016–2019. These definitions include a non-specified mixture of suspected and confirmed cases[4,54]. To estimate comparable case numbers in the period 2000–2008, when only DHF cases were available at a municipality level, we divided these case counts by the year-specific national proportion of all dengue cases that were classified as "dengue haemorrhagic fever" as reported by Dantés et al. [54]. For Brazil, case location was defined by the "estimated municipality of infection" while for Mexico the municipality in which the case was reported was assumed to be the municipality of infection. We assume the absence of reporting from a municipality in any particular year was indicative of the absence of dengue cases. Case counts were converted to incidence rates using the 2010 census population estimates for both countries.

### Climate data
A range of time-varying, gap-filled, remotely-sensed climate datasets generated by the Malaria Atlas Project[55] were downloaded via Google Earth Engine (https://developers.google.com/earth-engine/). These included annual mean and standard deviation of enhanced vegetation index (EVI), day and night land-surface temperature (LST), tasselled cap brightness (TCB) and tasselled cap wetness (TCW), and annual mean Landcover classification between the years 2000 and 2015. Each of these features has been shown previously to be experimentally and observationally associated with DENV transmission risk due to their influence on mosquito survival, population size, human contact rate, and DENV incubation rates[6,27]. All features were aggregated from their original 5 km × 5 km resolution to the municipality scale using population-weighted averages based on Worldpop 2015 UN-adjusted estimates[56]. Latitude, longitude, and distance between municipalities were calculated from the centroid of the municipality.

### Connectivity estimates
To measure the degree of connectedness between dengue-invaded source municipalities and potential vulnerable destinations we used seven different estimates of human mobility between municipalities that cover a spectrum of human movement types. Simple distance-based measures included: (1) great circle distance between municipality centroids and (2) municipality adjacency (binary yes/no). Movement model-based measures included (3) structured gravity and (4) radiation models[57] that take into account the influence of uneven population distributions. We also include quantitative measures of human movement with (5) land-surface travel time[58] to represent landscape and infrastructural heterogeneities, (6) disaggregated flight data from the global database of the International Air Transport Association (www.iata.org) and (7) between-state census migration data[59] to represent long-term movement flows which are also likely to be correlated with visiting friends and relative traffic. Further details on movement data sources and processing are available in Supplementary Information Section 1.1.

### Defining invasion
To classify municipalities as invaded or not invaded we explore a range of case count and incidence-based thresholds. We assume that invasion is non-reversible and that once a municipality is classified as invaded, it has the potential to seed invasions in other areas. Our definition of "invaded" encompasses endemic areas with year-round sustained DENV transmission, but also epidemic areas where frequent large autochthonous outbreaks pose a risk for seeding further spread even if they are not sustained locally over the long term. Our choice of threshold for defining invasion aimed to balance sensitivity and

specificity (i.e., detect a high number of invasion events, but also support the assumption that invaded municipalities continue to report ongoing transmission in post-invasion years). To optimise this threshold, we simulated a range of case and incidence-based thresholds and calculated: (i) the number of municipalities defined as invaded over the timeseries and (ii) the percentage of years in which dengue cases are reported in post-invasion years (Supplementary Fig. 1). The country-specific invasion threshold was chosen based on the value that lies closest to the most optimal point ([100,100], Supplementary Fig. 1).

### Constructing a spread dataset
Once invasion criteria were established, we constructed a spread dataset by separating municipalities into invaded or not-yet invaded categories by the end of each calendar year (Mexico: 1995–2019, Brazil 2001–2019). Year-specific environmental characteristics for each municipality were calculated and combined with year-specific metrics for each human movement type. Each human movement feature summarised the flux between each vulnerable municipality and its most closely connected DENV-invaded municipality with the list of invaded municipalities being updated annually. Collinearity between features was assessed by creating correlograms of Pearson correlations using Harrell's method[60], with a threshold of $p < 1 \times 10^{-6}$ to account for multiple hypothesis testing. Features with Pearson correlations greater than 0.75 (or less than −0.75) were considered for removal. Based on this process, four sets of collinear pairs emerged, with one feature in each pair removed from the final model. We removed latitude rather than the standard deviation of night-time land-surface temperature because meteorological features have a direct relationship with mosquito growth rates[27], Aedes suitability rather than mean night-time land-surface temperature due to the former not varying over time, great circle rather than friction surface movement due to the latter's closer link to human movement data[58], and finally gravity model movement rather than radiation model movement due to previous infectious disease spread studies that have suggested the superiority of radiation models due to their closer alignment with high-frequency movements[19,57]. All features were centred and scaled to have a mean of 0 and a standard deviation of 1 prior to model training and testing. A full list of features included in the final model is given in Fig. 3.

### Spread model structure
Our approach is designed to overcome three main challenges in modelling EID spread: (i) its drivers are multifactorial and complex, (ii) temporal label imbalance (few invaded at the beginning, few uninvaded by the end) and (iii) that spread is spatially conditional with the probability of invasion dependent on the invasion status of other areas. To address these we formulate a hierarchical spread model where we: (i) use machine-learning methods to capture non-linear and interacting feature effects[6] in a geospatial model that predicts annual invasion probabilities for each uninvaded municipality, (ii) fit an independent survival model (temporal model) to the national number of municipalities invaded to estimate the total number of invasions that should occur in any given year, and (iii) implement spread model prediction in a dynamic simulation framework where newly invaded areas and features are updated annually to capture the temporal dependence in the spread process.

The geospatial model aims to predict the probability of invasion of municipality $i$ based on climate features within municipality $i$ and connectivity to other invaded municipalities. We considered a wide range of supervised, statistical machine-learning models including: basic logistic regression (LR), penalised logistic regressions (lasso, ridge, and elastic net regression), k-nearest neighbours (KNN), decision tree (DT), random forest (RF), gradient-boosted decision trees (GBDT, XGBoost implementation), and a multilayer perceptron (MLP).

All models were developed using the *Tidymodels (v 0.1.0)* R package[61], which provides a unified framework for machine-learning workflows. To compare the relative performance of machine-learning methods for the geospatial model we first run a preliminary repeated naïve cross-validation of the spread data. We split municipality-year data-points for each country randomly into ten independent 75% training and 25% testing sets with no stratification for time or location. We tuned hyperparameters for each model by running a grid search over the hyperparameter space in combination with 5-fold cross-validation on the training set (Supplementary Information Section 1.2). Once optimal hyperparameters were found (i.e., those that maximised AUC over the 5-folds), we evaluated model performance on the held-out test sets. We considered three metrics commonly applied to classification problems: area under the receiver operating characteristic curve (AUC) to measure discrimination, sensitivity (SN), and specificity (SP). For the initial model comparison stage, we set the classification threshold by minimising the absolute value of the difference between SN and SP on the training set.

The temporal model aims to predict the total number of municipalities that are invaded in any given year. To estimate this we fitted a range of parametric survival models (exponential, Weibull, gamma, log-normal, Gompertz, log-logistic, and generalised gamma) and, for comparison, a non-parametric one-knot spline survival model to the annual number of municipalities invaded for each country using the *flexsurv* R package[62]. The spread model combines the geospatial and temporal sub-models by taking the annual invasion probabilities from the geospatial model, ranking them from highest to lowest then defining the top $n$ municipalities as invaded where $n$ is the annual number of expected invaded municipalities from the temporal model. To identify the optimal functional form for the temporal model at each time point we compared spread models with different functional forms over an expanding window timeseries cross-validation. The training data window began in 2003 in Brazil and 2000 in Mexico expanding until 2016 for both countries to ensure both training and testing datasets included at least three years of data. Predictive performance between models was assessed by root mean squared error between the predicted and observed number of invaded municipalities each year.

## Assessing the predictive performance of the combined spread model

The predictive performance of the combined spread model was assessed through three different cross-validation experiments that assessed different aspects of the models predictive ability[42,63]. In each of these, the geospatial and temporal models are re-fit to the training data (although machine-learning model selection and hyperparameter optimisation were not repeated). First a naïve 75% training 25% testing cross-validation was performed that split spread data randomly and did not account for the temporal dependence of the data but is more comparable to conventional machine-learning prediction approaches and provides a useful comparator for other cross-validation approaches. Second, a "hindcast" expanding window timeseries cross-validation fitting procedure was implemented. In this, we successively attempted to predict which municipalities would be invaded in year $t$ given the patterns of spread in all years before year $t$ and given which areas were invaded in year $t − 1$. This gradually expanded the amount of training data informing our model predictions and did not carry forward correct or incorrect predictions as the observed invasions status of each municipality was re-assigned each year. Finally, we ran a longer-term "simulation" timeseries cross-validation where the model was fit to all years of data for each country, but only initialised with the invaded area information at the beginning of the timeseries (2001 for Brazil, 1995 for Mexico). Step-wise annual predictions were then made up until 2019 with predicted invasion events carried forward year-to-year. This tested the ability of the model to reconstruct longer-term patterns of spread given a long-term average estimate of the factors driving spread and defined origins, and this approach was most relevant for testing the ability of the model to extrapolate to times outside the data date ranges. The simulation approach also allowed us to evaluate the year of arrival prediction residuals (i.e., year invaded − year predicted). We evaluated model performance across these setups in terms of year-over-year AUC, SN, and SP as well as visually with calibration and channel plots (Supplementary Fig. 4). A comparison between the naïve, hindcast, and simulation approaches evaluates the generalisability of the underlying spread process across different timescales.

To assess the generalisability of the fitted models for making predictions in different countries we conducted a between-country cross-validation using the simulation approach (i.e., trained on Mexico spread data and tested on Brazil spread data given the initial invaded municipalities in Brazil and vice versa).

## Interpreting the fitted model

To gain some insights into how different climate and connectivity features were linked to spread risk in the geospatial model we compute Shapley values[64,65] of the model fit to data from all years under the simulation approach. The Shapley values indicate how different values of features influence model predictions, but remain a simplification of their effect due to complicated interactions between features in an XGBoost model. To assess the relative role of climate and connectivity in driving dengue invasion, we calculate the combined contribution to model-predicted invasion risk of all movement and all environment features for each municipality in the year in which it was invaded.

To estimate the geographic sources of some key invasion events since 2000, we examined the human movement data at a more granular level for invasion events of Brasília (2010), São Paulo (2014), Monterrey (2005) and Guadalajara (Zapopan municipality, 2009). We included all connectivity features as the geospatial model fit a clear positive relationship with each feature (as assessed by Shapley values, Fig. 3). We then mapped the most connected municipality to the invaded area, as determined by each different human movement feature.

## Projecting historical spread in Brazil to identify the origins of the spread

Because dengue spread throughout Brazil before the establishment of nationwide surveillance the geographic origins of spread are unknown. To estimate patterns of dengue spread before the year 2000 we assembled a list of candidate geographic origins from phylogenetic analyses and outbreak reports. We then tested the likelihood of each candidate source by initialising our spread model with each source, simulating spread up to the year 2001, and then assessing correlation with the observed distribution of invaded municipalities in 2001.

To identify candidate locations and years of introduction, we examine a database of 8309 epidemiological records and 10,444 DENV genome sequences. Using a previously described dengue occurrence database[17], we identified early clusters of reported cases identified Rio de Janeiro (1986), Fortelaza (1986), Riberão Preto (1990), and Manaus (1996) as possible geographically distinct sources of sustained spread (Supplementary Fig. 7). To obtain estimates from genomic data we downloaded publicly available DENV genome sequences (≥8000 base pairs globally ≥1000 for Brazil) from GenBank/NCBI[66] and selected sequences that included country of origin (and region for Brazilian sequences) and year of collection ($n = 10,444$). Sequence alignments were performed with minimap2 v2.24[67] and gofasta v1.1.0[68]. An initial global maximum-likelihood (ML) tree was inferred and used to identify the clades corresponding to the main Brazilian DENV genotypes using IQTREE2[69], then ML phylogenies for each genotype were inferred separately. Visual inspection of these genotype ML trees identified nine main phylogenetic lineages for which we then estimated the

Article

spatiotemporal origins using a Bayesian phylogeographic framework in BEAST v.1.10.5[70]. All lineages with an origin date pre-2000 were considered as candidate origin sources. Further details on the identification of candidate sources from outbreak reports and phylogenetic analyses are available in Supplementary Information Section 1.3.

Each candidate source was then used to initialise the spread model (fit to all data under the "simulation" approach) which was then simulated up to the year 2001 and then evaluated. All combinations of candidate sources were also tested to represent scenarios where the modern spread of dengue in Brazil is the product of multiple independent introductions. The likelihood of different candidate sources and their combinations was expressed as an improvement in fit to the observed distribution in 2001 relative to a spread model initialised with the earliest candidate source (Rio de Janeiro 1983) as evaluated by AUC, sensitivity, specificity, positive predictive value and negative predictive value.

### Projecting future spread

For both countries, we projected the future spread until 2039. We initialised our models with the known invaded municipalities in 2019 and simulated the spread for each successive year. To account for changing mobility and climate features in the future and in the past when projecting historical spread we project values based on the national trend in the most relevant matched feature from either the Tier-1 CMIP6 future projection scenarios[37] for climate features or an equal weight of population and GDP trends for mobility features. Further details on this projection method are available in Supplementary Information Section 1.4 with summary outputs in Supplementary Fig. 8. To estimate uncertainty in the projected invasion date, we repeat the future simulations with low, medium, and high spread rate scenarios. These are based on the 97.5, 50 and 2.5 centile spread rate predictions from an ensemble of the predictions from the five temporal models fit to the most recent data in the timeseries cross-validation (i.e., 2001–2019, 2001–2018, 2001–2017, 2001–2016 and 2001–2015).

### Reporting summary

Further information on research design is available in the Nature Portfolio Reporting Summary linked to this article.

## Data availability

Dengue case count data is publicly accessible for Brazil from the Ministério da Saúde's DATASUS system (http://tabnet.datasus.gov.br/cgi/deftohtm.exe?sinanwin/cnv/denguebr.def, http://tabnet.datasus.gov.br/cgi/tabcgi.exe?sinannet/cnv/denguebr.def, http://tabnet.datasus.gov.br/cgi/deftohtm.exe?sinannet/cnv/denguebbr.def) and is available for Mexico from the Sistema de Vigilancia Epidemiológica de Dengue which can be accessed by contacting the Instituto Nacional de Transparencia (https://home.inai.org.mx). Processed versions of both datasets are provided in the study repository. Dengue virus sequence data was obtained from GenBank/NCBI (https://www.ncbi.nlm.nih.gov/genbank/). Climate, environmental and surface travel time covariates are freely available from the Malaria Atlas Project (https://data.malariaatlas.org/maps) and can be downloaded via Google Earth Engine (https://developers.google.com/earth-engine/). High-resolution population data and state-level migration estimates can be freely obtained from WorldPop (https://www.worldpop.org). Flight data is not freely available but can be purchased from IATA. Processed versions of all datasets used in analyses are provided in the study Figshare repository: https://doi.org/10.6084/m9.figshare.22047905.v2. Source data are provided with this paper.

## Code availability

All analyses were performed in the computing software R (version 4.0.1)[71] and Rstudio[72] (v2022.12.0) using the following packages: geosphere[73] (v1.5-18), spdep[74] (v1.2-7), tidyverse[75] (v1.3.0), Hmisc[60]

(v4.7-2), tmap[76] (v3.3), tidymodels[61] (v0.1.0), flexsurv[62] (v2.2.0), SHAPforxgboost[77] (v0.1.0). An archived version of all the code used for in this paper is available in the following Zenodo repository: https://doi.org/10.5281/zenodo.10890182. Larger processed variable files for environmental and human movement datasets have been deposited in the following Figshare repository: https://doi.org/10.6084/m9.figshare.22047905.v2.

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

## Acknowledgements

V.H. was supported through an Ontario Graduate Scholarship, Canadian Institutes of Health Research Banting and Best Master's and Doctoral Awards, and Vector Institute Postgraduate Affiliate Award. K.K. was supported through a Temerty Health Nexus Chair in Health Innovation and Technology at the University of Toronto. O.J.B. was supported by a UK Medical Research Council Career Development Award (MR/V031112/1). This project was supported by a Medical Research Council-São Paulo Research Foundation (FAPESP) CADDE partnership award (MR/S0195/1 and FAPESP 18/14389-0) (http://caddecentre.org/).

## Author contributions

Conceptualisation - V.H., O.J.B., S.I.H. Methodology - V.H., F.R.R.M., R.G., M.U.G.K., R.C.R., D.M.P., T.A.P., I.B., K.K., N.R.F., F.C.-M., F.C.-N., O.J.B. Formal analysis - V.H., F.J.C., F.R.R.M., N.R.F., O.J.B. Investigation - V.H., F.R.R.M., R.G., M.U.G.K., R.C.R., D.M.P., T.A.P., D.J.W., G.V.-P., P.M.S., G.L.B., E.C.S., N.R.F., F.C.-M., F.C.-N., O.J.B. Data curation - F.J.C., F.R.R.M., M.D., D.J.W., I.B., P.M.S., F.C.-M., F.C.-N. Writing original draft - V.H., O.J.B., S.I.H. Writing review and editing - all authors.

## Competing interests

K.K. is the founder and CEO of BlueDot, a B corporation that tracks emerging infectious diseases. M.D. is employed at BlueDot and V.H. was a paid intern at BlueDot in 2018. I.B. consults to BlueDot and to the NHL Players' Association. All other authors report no competing interests.

## Ethical approval

This research involved local researchers and government personnel in Brazil and Mexico to formulate research questions, identify relevant previous local research studies, interpret data, directly conduct the research and interpret the significance and utility of the findings, resulting in several individuals meeting the full criteria for authorship including leadership roles (F.R.R.M., P.M.S., G.L.B., E.C.S., F.C.-M. and F.C.-N.). The findings of this work are highly locally relevant in each country's ongoing efforts to prevent further spread of dengue in resource-limited settings. As part of the wider CADDE project (https://www.caddecentre.org) this work involved capacity building through workshops on specific technical skills and a wider commitment to deepening ongoing research collaborations with researchers at the University of São Paulo.

## Additional information

[1]Temerty Faculty of Medicine, University of Toronto, Toronto, ON, Canada. [2]Dalla Lana School of Public Health, University of Toronto, Toronto, ON, Canada. [3]Vector Institute for Artificial Intelligence, Toronto, ON, Canada. [4]Centre for the Mathematical Modelling of Infectious Diseases, London School of Hygiene & Tropical Medicine, London, UK. [5]Department of Infectious Disease Epidemiology, Faculty of Epidemiology and Population Health, London School of Hygiene & Tropical Medicine, London, UK. [6]Centre on Climate Change and Planetary Health, London School of Hygiene & Tropical Medicine, London, UK. [7]Medical Research Council Centre for Global Infectious Disease Analysis, Abdul Latif Jameel Institute for Disease and Emergency Analytics and Department of Infectious Disease Epidemiology, School of Public Health, Imperial College London, London, UK. [8]Departamento de Genética, Universidade Federal do Rio de Janeiro, Rio de Janeiro, RJ, Brazil. [9]Department of Genetics, Evolution and Environment, University College London, London, UK. [10]Department of Biology,

University of Oxford, Oxford, UK. [11]BlueDot, Toronto, ON, Canada. [12]Institute for Health Metrics and Evaluation, University of Washington, Seattle, WA, USA. [13]Department of Health Metrics Sciences, School of Medicine, University of Washington, Seattle, WA, USA. [14]Department of Biological Sciences, University of Notre Dame, Notre Dame, IN, USA. [15]Eck Institute for Global Health, University of Notre Dame, Notre Dame, IN, USA. [16]Geospatial Health and Development, Telethon Kids Institute, Nedlands, WA, Australia. [17]Faculty of Health Sciences, Curtin University, Perth, WA, Australia. [18]Divisions of General Internal Medicine and Infectious Diseases, Toronto General Hospital, University Health Network, Toronto, ON, Canada. [19]Department of Environmental Sciences, Emory University, Atlanta, GA, USA. [20]Autonomous University of Yucatan, Merida, YUC, Mexico. [21]Pasteur Institute, State Secretary of Health of São Paulo, São Paulo, SP, Brazil. [22]Institute of Tropical Medicine, Faculdade de Medicina, Universidade de São Paulo, São Paulo, SP, Brazil. [23]Division of Infectious Diseases, St. Michael's Hospital, Unity Health Toronto, Toronto, ON, Canada. [24]Li Ka Shing Knowledge Institute, Unity Health Toronto, Toronto, ON, Canada. [25]Centro Nacional de Programas Preventivos y Control de Enfermedades (CENAPRECE) Secretaria de Salud Mexico, Ciudad de Mexico, Mexico. [26]Department of Epidemiology, School of Public Health, University of São Paulo, São Paulo, SP, Brazil. ✉e-mail: oliver.brady@lshtm.ac.uk

