## [Peer Review File · Nature Communications]

REVIEWERS' COMMENTS

Reviewer #1 (Remarks to the Author):

The authors have clarified a number of points that are helpful to understanding their methodology and interpreting their results.

Something that I feel is still missing from the main text is an acknowledgment that the reconstructed and predicted rates of spread used throughout the results are solely driven by observed rates of invasion using the survival model and do not depend on any connectivity or environmental covariates. In its current form, it can be easily misinterpreted that the environmental and connectivity factors are being used to predict rates of invasion which is not the case.

I find the results section "Mapping routes of spread" to be quite hard to follow in relation to the Figures. The results of some panels of Figures 4 and 5 are never mentioned in the text. For instance the authors show very specific routes with exact sources of future DENV invasion for major cities in the figures but these are not mentioned in the text.

The discussion section seems to be entirely focused on all the things the authors say they can do with this framework. A summary of key findings, novelty, etc., might help to lead up to the justification for this.

Reviewer #3 (Remarks to the Author):

In my opinion, the authors did a great job addressing several important concerns raised by the reviewers. My primary expertise is not modeling, so I cannot directly assess all of the responses to reviewers 1 and 2. My concerns about the phylogenetic availability, the framework for outside usage, and questions about the model have been adequately addresses. I have no further concerns.

Reviewer #1:

The authors have clarified a number of points that are helpful to understanding their methodology and interpreting their results.

Something that I feel is still missing from the main text is an acknowledgment that the reconstructed and predicted rates of spread used throughout the results are solely driven by observed rates of invasion using the survival model and do not depend on any connectivity or environmental covariates. In its current form, it can be easily misinterpreted that the environmental and connectivity factors are being used to predict rates of invasion which is not the case.

We thank the reviewer for bringing this lack of clarity to our attention. In response we have made several changes (in *red italics*) to the main text where this is presented. First, in the methods summary (around line 156, results section) we now state:

“Our two-fold approach first uses a hierarchical survival (temporal) model to predict the total number of municipalities invaded each year *without any connectivity or environmental covariates*. Second, a machine learning (geospatial) model trained on year-on-year changes in invasion sources and a range of environmental and connectivity features determines the spatial distribution of invaded municipalities each year (Methods).”

And in the section where the results of the survival model are specifically discussed we now state:

“Despite important heterogeneities, the annual total number of invaded municipalities was well represented by *covariate-free* parametric survival models that gave better predictive performance than more flexible spline-based approaches (SI Fig. 2A-B).”

I find the results section “Mapping routes of spread” to be quite hard to follow in relation to the Figures. The results of some panels of Figures 4 and 5 are never mentioned in the text. For instance the authors show very specific routes with exact sources of future DENV invasion for major cities in the figures but these are not mentioned in the text.

We thank the reviewer for identifying this lack of clarity and have made some minor changes to the sections where these results are discussed to improve clarity. The “Mapping routes of spread” intentionally only describes the results of past invasion events (Fig. 4A-B and Fig. 5A-B) while the future spread predictions (Fig. 4C-D and Fig. 5C-D) are described in the “Predicting future spread between 2020 and 2039” section. For space efficiency they are presented as combined figures. For brevity the text in the results focusses on common patterns across these different invasion events while the figures (and new source data) provide additional details should the reader be interested in the estimates for specific cities. Nonetheless, we have made a series of changes to add detail and make the links between the text and specific figures more clear.

In the mapping route of spread section we have made the following changes:

“Closer examination of the mobility networks associated with observed invasion of the largest cities (Fig. 4-5) suggest that invasion is a multi-stage process. For Monterrey, Guadalajara, Brasília, and São Paulo, multiple lower-density neighbouring areas or nearby cities were invaded in the years before the city-centre itself. In all cases, connectivity via long distance air routes was present for many years before invasion occurred. *In Brazil, Rio de Janeiro (invaded in or before 2001) showed a high degree of air and migration connectivity with both Brasília and São Paulo but invasion did not occur until 2010 and 2014 respectively (Fig. 4A-B), with Cancún (invaded since 2001) playing a similar role in Mexico (Fig. 5A-B).* This would suggest that for major cities to be invaded, connectivity by air is necessary (as also demonstrated by the geospatial model (Fig. 3A and D)) but not sufficient for invasion which instead requires the combined importation pressure of both nearby and longer distance links (Fig. 4-5).”

In the “Future routes of spread” section we now state:

“This will include spread to the last few remaining dengue-free large cities in Mexico with the areas around Tijuana, bordering the USA in the far north, expected to be invaded between 2027 and 2030 and the first invasions into metropolitan Mexico City between 2038-2039 (Fig. 5C-D). *Invasion into Tijuana is expected as a continuation of a gradual spread up the gulf of California with the highest invasion risk from the neighbouring municipalities of Ensenada and Mexicali as well as the more distance regional centres of La Paz and Guadalajara (Fig. 5C). The first invasions into the Mexico City metropolitan area are expected to occur when longer distance connectivity (Cancún) combines with links to other more proximal regional cities (Aguascalientes and Puebla, Fig. 5D). In Brazil, the majority (60.7%) of areas invaded 2020-2039 will be in the South region with the more isolated areas of the state of Santa Catarina and Rio Grande do Sul being the last to be invaded (Fig. 7C-D). With strong connections with São Paulo in combination with a gradual advance of dengue down the southern Brazilian coastline, the two biggest cities in southern Brazil (Porto Alegre and Curitiba) are expected to be invaded in 2022 (CI 2022-2022) and 2024 (CI 2024-2025) respectively (Fig. 4C-D)*”

The discussion section seems to be entirely focused on all the things the authors say they can do with this framework. A summary of key findings, novelty, etc., might help to lead up to the justification for this.

We agree with the reviewer that the discussion would benefit from a summary of the work’s findings and their novelty. The first paragraph of the discussion has been re-written to reflect this:

“*Our analyses showed that the expansion of dengue in Mexico and Brazil follows consistent and predictable pathways that are shaped by an interaction between environmental suitability of the destination and connectivity with potential sources. By using models that account for these drivers we showed that the modern spread of dengue in Brazil could be explained by just three introductions to Rio de Janeiro, Fortaleza and Manaus between 1983-1996, identify likely proximal and distal routes of invasion for specific cities and project the timing of future spread into highland regions of Mexico, including Mexico City, and southern*

Brazil. This represents the first time, to our knowledge, that spatial models of disease spread have informed origins, pathways, and future projections of an emerging infectious disease."

Reviewer #3:

In my opinion, the authors did a great job addressing several important concerns raised by the reviewers. My primary expertise is not modeling, so I cannot directly assess all of the responses to reviewers 1 and 2. My concerns about the phylogenetic availability, the framework for outside usage, and questions about the model have been adequately addresses. I have no further concerns.

We thank the reviewer for acknowledging the improvements we have been able to make to this work as a result of the combined reviewer comments and are glad that no issues remain.